# Tracing In-Hospital COVID-19 Outcomes: A Multistate Model Exploration (TRACE)

**DOI:** 10.3390/life14091195

**Published:** 2024-09-21

**Authors:** Hamed Mohammadi, Hamid Reza Marateb, Mohammadreza Momenzadeh, Martin Wolkewitz, Manuel Rubio-Rivas

**Affiliations:** 1Biomedical Engineering Department, Engineering Faculty, University of Isfahan, Isfahan 81746-73441, Iran; bme.mohammadi@eng.ui.ac.ir; 2Department of Automatic Control (ESAII), Biomedical Engineering Research Centre (CREB), Universitat Politèncicna de Catalunya (UPC), 08028 Barcelona, Spain; 3Department of Artificial Intelligence, Smart University of Medical Sciences, Tehran 1553-1, Iran; momenzadeh.m@smums.ac.ir; 4Institute of Medical Biometry and Statistics, Faculty of Medicine and Medical Center, University of Freiburg, 79104 Freiburg, Germany; martin.wolkewitz@uniklinik-freiburg.de; 5Department of Internal Medicine, Bellvitge University Hospital, Hospitalet de Llobregat, 08907 Barcelona, Spain; mrubio@bellvitgehospital.cat

**Keywords:** COVID-19, comorbidity, diabetes mellitus, hospital length of stay, lymphocytes, mortality, multistate model, prognosis, risk factors

## Abstract

This study aims to develop and apply multistate models to estimate, forecast, and manage hospital length of stay during the COVID-19 epidemic without using any external packages. Data from Bellvitge University Hospital in Barcelona, Spain, were analyzed, involving 2285 hospitalized COVID-19 patients with moderate to severe conditions. The implemented multistate model includes transition probabilities and risk rates calculated from transitions between defined states, such as admission, ICU transfer, discharge, and death. In addition to examining key factors like age and gender, diabetes, lymphocyte count, comorbidity burden, symptom duration, and different COVID-19 waves were analyzed. Based on the model, patients hospitalized stay an average of 11.90 days before discharge, 2.84 days before moving to the ICU, or 34.21 days before death. ICU patients remain for about 24.08 days, with subsequent stays of 124.30 days before discharge and 35.44 days before death. These results highlight hospital stays’ varying durations and trajectories, providing critical insights into patient flow and healthcare resource utilization. Additionally, it can predict ICU peak loads for specific subgroups, aiding in preparedness. Future work will integrate the developed code into the hospital’s Health Information System (HIS) following ISO 13606 EHR standards and implement recursive methods to enhance the model’s efficiency and accuracy.

## 1. Introduction

Infectious diseases caused by microorganisms that invade the body can range from mild to severe and are transmitted directly and indirectly [1]. Despite advances in controlling many infections, humans remain vulnerable to various illnesses [2]. Coronavirus disease is caused by the acute respiratory syndrome coronavirus 2 (SARS-CoV-2) [3]. Coronavirus Disease 2019 (COVID-19) rapidly spread worldwide, resulting in over seven million deaths. The World Health Organization declared it a global pandemic on 11 March 2020. By 31 December 2023, WHO reported over 7.7 billion infections and more than 7 million deaths [4]. In 2020, COVID-19 became the third leading cause of death, mainly affecting older adults and those with underlying conditions. CDC data from 2020 to 2022 highlighted that unvaccinated individuals aged 50 and older faced a significantly higher risk of death than those under 30 [5].

Multi-state models offer a flexible framework for describing a person’s life history in relation to time and events. They are defined as random processes and are well-suited for analyzing longitudinal data or repeated events [6,7,8].

Hospital costs are closely tied to the length of stay, which can vary widely depending on patient characteristics, social conditions, and treatment complexity [9]. Shorter stays generally lead to better outcomes, including reduced infection risks and mortality, while lowering costs [10,11]. As a result, practical tools are needed to identify at-risk patients and optimize care quickly.

The investigations carried out on multi-state models for the analysis of survival in COVID-19 are as follows:

Hung et al. conducted a study investigating the relationship between right ventricular (RV) phenotypes and mortality in ICU patients with COVID-19-induced ARDS. Their analysis showed that RV involvement fluctuated, with patients displaying acute cor pulmonale in the final echocardiographic exam facing the highest mortality risk [12]. Gonzalez et al. explored the role of neutrophil extracellular traps (NETs) and platelet activation in predicting clinical outcomes in COVID-19 patients. Their study of 204 patients, categorized by clinical severity, found that elevated cell-free DNA (cfDNA), citrullinated histone H3 (CitH3), and P-selectin levels were linked to more severe outcomes. Notably, higher CitH3 and P-selectin levels were associated with an increased risk of ICU transfer and death [13]. Edmondson et al. analyzed the length of stay (LOS) in the emergency department for patients with substance use disorder, using a multi-state model to identify factors influencing the time spent in different stages of care and the key variables affecting the duration of each state [14].

Hazard et al. utilized multi-state models to analyze the clinical progression of hospitalized COVID-19 patients, particularly in cases of severe pneumonia requiring intensive care, ventilation, or oxygenation. Their study provided estimates of the risks and probabilities of transitions between clinical states, helping to quantify the average duration patients spent in critical care settings like ICUs and on invasive ventilation [15]. Jarrett et al. conducted a cohort study to assess the impact of air pollution on COVID-19 severity and recovery by analyzing individual patient data. They evaluated outcomes like ICU worsening, death, recovery, and discharge followed by death, using a multi-state model to estimate transmission risks [16]. Relan et al. investigated oxygen use requirements in COVID-19 patients, focusing on the type and duration of different oxygen therapies. Using a multi-state model, they analyzed hospitalization periods, examining transitions between increasing levels of respiratory support and recovery. Their study, conducted on a hospital cohort, also explored COVID-19 patients with cardiovascular complications [17].

Abbas et al. conducted a retrospective cohort study using a fully parametric multistate model within a Bayesian framework to estimate the competing risks of death and discharge among hospitalized COVID-19 patients and uninfected patients [18]. Amdid et al. analyzed the clinical progression of COVID-19 with a six-state model and Poisson regression, focusing on factors like diabetes duration, age, and the impact of diabetes-related complications and drugs [19]. Shaw et al. performed a cohort analysis on PCR-confirmed COVID-19 patients, using multivariate Cox regression to study transmission and outcomes such as discharge, readmission, and mortality [20]. A study in Milan used a multi-state model to evaluate COVID-19 patients over time, estimating probabilities and timing of different events [21]. Mehri et al. identified significant associations between COVID-19 mortality and various clinical and laboratory factors through their systematic review and meta-analysis. Specifically, an increased white blood cell (WBC) count (Odds Ratio (OR) = 1.42, *p*-value = 0.023), decreased lymphocyte levels (OR = 1.68, *p*-value < 0.001), elevated blood urea nitrogen (BUN) (OR = 1.05, *p*-value = 0.046), increased creatinine (OR = 1.82, *p*-value = 0.001), and vitamin D deficiency (OR = 2.64, *p*-value < 0.001) were significantly associated with higher mortality. Additionally, comorbidities such as diabetes (OR = 1.76, *p*-value < 0.001), hypertension (OR = 1.64, *p*-value = 0.002), cardiovascular disease (CVD) (OR = 1.74, *p*-value < 0.001), and chronic kidney disease (CKD) (OR = 2.27, *p*-value < 0.001) were also found to be significantly linked to increased mortality from COVID-19 [22]. These findings emphasize the need for effective hospital capacity planning to address increased demand for ICU beds and ventilators.

The COVID-19 pandemic has created significant challenges, including increased demand for hospital and ICU beds and ventilators, leading to overcrowding and reduced capacity. Effective planning based on individual patient records and future projections is crucial for managing hospital capacity and tracking patient trends. Additionally, the territorial decentralization of health policy complicates epidemic management, as divided healthcare services often result in inefficiencies [23]. The pandemic has highlighted how decentralization, driven by political rather than managerial criteria, can disrupt the healthcare system’s effectiveness [24].

Spain’s healthcare system is grappling with an epidemic crisis intensified by cost containment, instability among health personnel, and structural challenges [25]. A study on the duration of hospitalization for COVID-19 patients showed that hospital stays ranged from 2 to 10 days in the United States, 1 to 6 days in Italy, and 5 to 19 days in Germany, with an average of 6 days in Spanish hospitals [26]. The spread of COVID-19 has posed a significant challenge, causing substantial damage in Spain [27].

The existing literature on hospital length of stay during the COVID-19 epidemic largely relies on multistate models implemented through external packages, particularly in R. However, this approach often requires specific toolboxes and limits flexibility in embedding these models into diverse healthcare systems. In contrast, our study fills this gap by developing a multistate model without external packages or toolboxes, making it easily adaptable to various environments, including hospital dashboards. This study aims to estimate, forecast, and manage hospital length of stay during the COVID-19 epidemic using a custom-implemented multistate model applicable across platforms without additional software. We focus on a hospital in Spain, detailing the data analysis, the steps involved in our model’s development and estimation, and the results achieved. This approach enhances accessibility and practicality for healthcare providers, offering a straightforward integration into hospital management systems.

## 2. Materials and Methods

### 2.1. Materials

We analyzed data from 2285 hospitalized COVID-19 patients at Bellvitge University Hospital in Barcelona, Spain, to implement and evaluate the 4-state model proposed in this study. These patients admitted between March 2020 and February 2021 experienced outcomes such as in-hospital mortality, discharge, or transfer to other medical facilities. The study period was categorized into three pandemic waves in Spain [28]:First period: From the beginning of the pandemic until 21 June 2020.Second period: From 22 June to 6 December 2020.Third period: From 7 December 2020 to 14 March 2021.

To be included in this study, patients must be 18 years or older, have SARS-CoV-2 infection confirmed by Reverse Transcription Polymerase Chain Reaction (RT-PCR), and have not received therapeutic interventions prior to hospitalization. Key variables considered were age, gender, Charlson Comorbidity Index (categorized as 0 and greater than 0), and inflammatory markers such as C-reactive protein (mg/L), lactate dehydrogenase (U/L), D-dimer (ng/mL), and lymphocytes (×10^6^/L). The follow-up began at the time of hospital admission (time zero) and lasted up to 45 days [29,30]. A representative dataset is available in Appendix A.

### 2.2. Methods

This section calculates the transition matrix, transition probabilities, and their related relationships to implement the 4-state model. The flowchart associated with the 4-state model is shown below.

First, the transition matrix is calculated from the available data, illustrating the transitions between different states. This matrix quantifies the number of transitions, with rows representing the ‘from’ state and columns the ‘to’ state. Entries with no recorded transitions remain zero. Based on previous studies [31], the transition probability for the 4-state model can be derived by calculating the transition intensity, which reflects the average number of transitions per unit of time (e.g., every three units).

For the proposed 4-state model, these transition intensities are determined using the following matrix:(1)$$\left[\begin{array}{cccc} -(q_{12}+q_{13}+q_{14}) & q_{12} & q_{13} & q_{14}\\ q_{21} & -(q_{21}+q_{23}+q_{24}) & -q_{23} & q_{24}\\ 0 & 0 & 0 & 0\\ 0 & 0 & 0 & 0 \end{array}\right]$$
where $q_{ij}$ represents the transition from state *i* to state *j*; these indices are defined in Figure 1.

Furthermore, to calculate the probability of transition based on the transition intensity, the following relationships apply:(2)$$P_{11}\left(t\right)=e^{-(q_{12}+q_{13}+q_{14})}\;P_{12}(t)=\frac{q_{12}}{\left(\left(q_{21}+q_{23}+q_{24}\right)-(q_{12}+q_{13}+q_{14})\right)}(e^{-(q_{12}+q_{13}+q_{14})t}-e^{-\left(q_{21}+q_{23}+q_{24}\right)t})\;P_{13}(t)=\frac{q_{13}}{q_{12}+q_{13}+q_{14}}(1-e^{-(q_{12}+q_{13}+q_{14})t})\;P_{14}\left(t\right)=\frac{q_{14}}{q_{12}+q_{13}+q_{14}}\left(1-e^{-\left(q_{12}+q_{13}+q_{14}\right)t}\right)=1-(P_{11}\left(t\right)+P_{12}\left(t\right)+P_{13}\left(t\right)\;P_{21}(t)=\frac{q_{21}}{\left(\left(q_{13}+q_{12}+q_{14}\right)-(q_{21}+q_{23}+q_{24})\right)}(e^{-(q_{21}+q_{23}+q_{24})t}-e^{-\left(q_{13}+q_{12}+q_{14}\right)t})\;P_{22}\left(t\right)=e^{-\left(q_{21}+q_{23}+q_{24}\right)t}\;P_{23}(t)=\frac{q_{23}}{q_{21}+q_{23}+q_{24}}(1-e^{-(q_{21}+q_{23}+q_{24})t})\;P_{24}\left(t\right)=1-(P_{21}\left(t\right)+P_{22}\left(t\right)+P_{23}\left(t\right))$$
where, $P_{ij}$ represents the probability of transition from state *i* to state *j*.

The matrix illustrating the related relationships can be displayed as follows:(3)$$\left[\begin{array}{cccc} P_{11}(t) & P_{12}(t) & P_{13}(t) & P_{14}(t)\\ P_{21}(t) & P_{22}(t) & P_{23}(t) & P_{24}(t)\\ P_{31}(t) & P_{32}(t) & P_{33}(t) & P_{34}(t)\\ P_{41}(t) & P_{42}(t) & P_{43}(t) & P_{44}(t) \end{array}\right]$$

This matrix displays the transition probabilities from one state to another. The code related to the above items is provided in Appendix A.

Figure 2 shows the flowchart of the implementation steps based on the written programming code.

### 2.3. Statistical Analysis

#### 2.3.1. Sub-Group Analysis

In this study, multistate analysis was applied to both the entire dataset and various sub-groups. Given the rapid spread of the coronavirus and evidence suggesting that increasing age leads to distinct physiological and pathological changes during infection [32], one analysis focused on age, categorizing patients into the following groups: less than 45 years, 45 to 54 years, 55 to 64 years, 65 to 74 years, and over 74 years [33]. Additionally, research has shown gender differences and the impact of diabetes on immunity and response to infectious diseases in both women and men. Therefore, another analysis was conducted based on gender and the presence of diabetes to explore these factors [34,35].

One sub-group focused on reducing lymphocytes, a key laboratory parameter often observed in COVID-19 infections, as it plays an essential role in disease prognosis [34]. Another sub-group analysis was conducted based on Charlson’s Comorbidity Index (CCI), a widely recognized tool for assessing survival rates in individuals with underlying diseases. For both CCI and lymphocyte levels, cutoff values were determined using the Youden Index to maximize sensitivity and specificity. The Youden Index (J) is an effective measure of a diagnostic test’s performance, helping to accurately classify outcomes by evaluating the balance between sensitivity (correctly identifying deaths) and specificity (correctly identifying discharges).

In addition to the previously mentioned sub-group analyses, we created a grouping based on the time between the onset of symptoms and the patient’s admission to the hospital, referred to as symptom duration. We calculated the median symptom duration and divided the patients into two sub-groups: those with less than the median duration and those greater than or equal to the median. This categorization enabled us to assess how the timing of medical attention influenced disease progression and patient outcomes.

#### 2.3.2. Statistical Methods

We employed several statistical methods to assess the differences between discharged patients and those who experienced in-hospital mortality. Continuous variables, such as age and lymphocyte count, were summarized using mean, standard deviation (SD), and median with minimum and maximum values. Categorical variables, including gender, diabetes status, and the Charlson Comorbidity Index, were summarized as counts and percentages. The statistical significance of differences between the two groups (discharged/death) was evaluated using the chi-square test for categorical variables. The Shapiro–Wilk test was used to assess the normality of continuous variables. For variables with a normal distribution, comparisons between groups were made using the independent *t*-test. The Mann–Whitney U test was applied to variables that did not follow a normal distribution. We used the Chi-square test to identify significant associations between categorical variables.

In this study, we assessed the distribution and relationships between symptom duration and length of stay (LOS) for COVID-19 patients across different pandemic waves using various statistical methods. The median of the symptom duration was also calculated, and the dichotomized symptom duration was also analyzed. Given the non-normal distribution of the variables, Spearman’s rank-order correlation was employed to evaluate the relationship between symptom duration and LOS. Additionally, a Pearson Chi-Square test was used to assess the association between categorical variables related to COVID-19 outcomes and the different waves. The Kruskal–Wallis test was applied to compare LOS across the three waves, with post hoc pairwise comparisons adjusted using the Bonferroni correction for multiple comparisons. The statistical analysis was conducted using IBM SPSS Statistics for Windows, Version 29.0 (Released 2022; IBM Corp., Armonk, NY, USA). Multi-state modeling was performed using MATLAB version 9.12 (R2022b) (Released 2022; The MathWorks Inc., Natick, MA, USA). A *p*-value of less than 0.05 was considered statistically significant.

## 3. Results

Table 1 summarizes the key characteristics and outcomes of the study population, including age, gender, diabetes status, lymphocyte count, and Charlson Comorbidity Index, comparing patients who were discharged with those who experienced in-hospital mortality. The *p*-values indicate the statistical significance of the differences observed between the groups.

CCI categories of (0 vs. >1) were defined by maximizing Youden’s J-Index. The related sensitivity, specificity, and AUC were 70% [CI 95: 65–74%], 55% [CI 95%: 53–57%], and 0.62 [CI 95%: 0.59–0.65], respectively. Such indices for lymphocyte categories (>910 × 10^6^/L vs. <910 × 10^6^/L) were 55% [CI 95%: 52–57%], 68% [CI 95%: 64–72%], and 0.61 [CI 95%: 0.59–0.64], respectively. The receiver operating characteristic (ROC) plots for CCI and Lymphocyte, including the CI 95% plots, are provided in Appendix A.

Moreover, the dataset used in this study was divided according to the defined COVID-19 waves for sub-group analysis, showing that 61% of the data correspond to the first wave, 19% to the second wave, and 20% to the third wave.

Spearman’s correlation revealed a weak but significant negative relationship between symptom duration and LOS (r = −0.094, *p*-value < 0.001). Moreover, the dichotomized symptom duration (<7 vs. ≥7 days) was weakly associated with the outcome variable (death/discharge) (*p*-value < 0.001; Phi = −0.113). The Pearson Chi-Square test showed no significant association between the outcome variable (death/discharge) and the COVID-19 waves (*p*-value = 0.320). The Kruskal–Wallis test identified significant differences in LOS across the three waves (*p*-value < 0.001). In the pairwise comparisons, LOS was significantly higher in Wave 2 compared to Wave 1 (adjusted *p* < 0.001) and also significantly higher in Wave 3 compared to Wave 1 (adjusted *p*-value <0.001), while there was no significant difference between Wave 2 and Wave 3 (adjusted *p*-value = 1.000).

Based on the multistate model analysis of COVID-19 data, the estimated length of stay (LOS) for each transition between states is as follows: patients hospitalized stay an average of 11.90 days before being discharged, 2.84 days before moving to the ICU, or 34.21 days before death. Patients in the ICU remain there for about 24.08 days on average, with subsequent stays of 124.30 days before discharge and 35.44 days before death. These results highlight the varying durations and trajectories of hospital stays for COVID-19 patients, providing critical insights into patient flow and healthcare resource utilization.

The Results section presents a series of stacked probability plots derived from the 4-state model for in-hospital COVID-19 patients. These figures provide a comprehensive visual representation of the transitions between different states of patient outcomes over time. The analysis includes the overall patient population and various subgroups to highlight differences based on specific characteristics.

We first provide the data related to the time of onset of symptoms and the admission time, as illustrated in Figure 3.

Figure 4 presents the output of the four-state model for the complete dataset of hospitalized COVID-19 patients, capturing the probabilities of transitioning between different states, including hospitalization, discharge, transfer to other facilities, and in-hospital mortality.

Figure 5 breaks down the four-state model by gender, displaying separate plots for (a) female and (b) male patients. This subgroup analysis elucidates potential gender-based differences in patient outcomes and transition probabilities. Figure 6 presents the four-state model outputs stratified by age groups, revealing how the probabilities of different outcomes vary across age categories. Figure 7 focuses on the impact of diabetes status on patient outcomes, showing separate plots for (a) non-diabetic and (b) diabetic patients, highlighting the differences in the transition probabilities between these groups.

Figure 8 presents the four-state model results based on lymphocyte counts, with plots for (a) patients with lymphocyte counts greater than 910 and (b) patients with lymphocyte counts less than 910. This analysis highlights the importance of lymphocyte levels as a key laboratory parameter in predicting patient outcomes. Figure 9 explores the four-state model outcomes according to the Charlson Comorbidity Index (CCI), showing plots for (a) patients with a CCI of 0 and (b) patients with a CCI greater than 0, illustrating the impact of underlying comorbidities on the progression of COVID-19 in hospitalized patients.

Finally, the results of dataset selection based on the different COVID-19 waves are presented in Figure 10.

These figures collectively provide a detailed overview of the dynamics and heterogeneity in patient outcomes, facilitating a deeper understanding of the factors influencing the prognosis of hospitalized COVID-19 patients.

## 4. Discussion

The stacked probability plot generated by the multi-state model clearly visualizes the evolving probabilities of patients transitioning between different health states (hospital admission, ICU, discharge, and death) over time. This graphical representation effectively captures how, at any given time, a patient is likely to be in one of these states. The model calculates these transition probabilities based on an empirically derived transition matrix built from observed patient data. The matrix exponentiation ensures that the probabilities of staying in or moving between states are accurately represented at each time point.

Additionally, the model allows for incorporating specific patient subgroups by adjusting the transition matrix according to relevant covariates, such as age or comorbidity index, making the stacked probability plot a flexible tool for understanding the heterogeneity in patient outcomes. This visualization benefits clinicians and healthcare planners as it helps assess hospital resource needs over time, such as ICU occupancy or discharge rates, and could be used to predict the expected patient flow during future pandemic waves or similar public health crises.

The analysis of death probabilities across different subgroups using the four-state model reveals significant insights into the factors influencing in-hospital mortality among COVID-19 patients.

The overall death probability in the patient cohort highlights the severity of COVID-19 among hospitalized individuals (Figure 4). The model indicates that a considerable portion of patients transitioned to mortality, underscoring the critical need for effective clinical interventions and healthcare resources to manage severe cases. This finding aligns with other studies that have shown high mortality rates among hospitalized COVID-19 patients, emphasizing the importance of early and aggressive treatment strategies.

The gender-specific analysis shows distinct differences in death probabilities between male and female patients. Male patients exhibited a higher probability of death compared to females (Figure 5). This finding is consistent with the existing literature suggesting that men may have a higher risk of severe outcomes from COVID-19 due to differences in immune response and comorbid conditions. Studies have also indicated that hormonal differences and lifestyle factors may contribute to this disparity [35,36].

The age-stratified plots reveal a clear trend: death probability increases with age. Patients over 75 years (B4) showed the highest probability of mortality, while those younger than 45 years (B0) had the lowest (Figure 6). This trend reflects the greater vulnerability of older adults to severe COVID-19 outcomes, likely due to a combination of weaker immune responses and a higher prevalence of underlying health conditions in older populations. Similar findings have been reported in other studies, highlighting age as a critical factor in COVID-19 mortality [37].

The subgroup analysis based on diabetes status indicates that diabetic patients have a higher probability of death than non-diabetic patients (Figure 7). This aligns with the understanding that diabetes is a significant risk factor for severe COVID-19, possibly due to impaired immune function and increased likelihood of other comorbidities [38]. Previous research has also demonstrated that diabetes can exacerbate the severity of viral infections, further increasing mortality risk [39].

Patients with lymphocyte counts less than 900 had a higher probability of death compared to those with counts greater than 900 (Figure 8). A low lymphocyte count is associated with a poorer prognosis in COVID-19 patients, reflecting the importance of lymphocyte levels as a marker for immune competence and disease severity. This finding supports the use of lymphocyte count as a prognostic indicator in clinical settings [40,41].

The analysis based on the Charlson Comorbidity Index (CCI) shows that patients with a CCI greater than 0 had a significantly higher probability of death compared to those with a CCI of 0 (Figure 9). This highlights the impact of pre-existing comorbidities on COVID-19 outcomes, with higher CCI scores indicating a greater burden of chronic illnesses that exacerbate the risk of mortality. These findings underscore the need for tailored care strategies for patients with multiple comorbidities [42,43].

The association between COVID-19 waves, symptom duration, and patient outcomes in our study provides critical insights into how the timing of disease progression and external pandemic factors influenced patient prognosis. The data revealed that the length of stay (LOS) significantly varied across the three COVID-19 waves, with patients in the second and third waves experiencing longer hospitalizations compared to those in the first wave. This may reflect changes in the clinical management of COVID-19, as later waves saw more severe cases due to the emergence of new variants and the adjustment of treatment protocols as healthcare systems adapted to the pandemic. Additionally, the dichotomized symptom duration (<7 vs. ≥7 days) was weakly associated with the outcome variable (death/discharge). However, the analysis showed no significant association between COVID-19 waves and patient outcomes (*p* = 0.320), indicating that while LOS differed across waves, the mortality risk was not directly tied to specific pandemic periods.

One limitation of the method presented in this study is its reliance on data from a single hospital, which may limit the generalizability of the results to other settings with different healthcare infrastructures or patient demographics. Additionally, the multistate model, while flexible, assumes that the transition intensities between states remain constant over time, which may not fully capture the dynamic nature of disease progression, especially in the context of rapidly evolving clinical practices during the COVID-19 pandemic. Finally, some key variables, such as vaccination status, were not included in the analysis, potentially impacting the accuracy of the model’s predictions.

## 5. Conclusions

The multistate model’s analysis reveals that age, low lymphocyte count, and a higher Charlson Comorbidity Index (CCI) significantly influenced the transition probabilities between different health states in COVID-19 patients. Interestingly, while diabetes diagnosis alone did not show a substantial effect in subgroup analysis, the importance of CCI suggests that the overall burden of comorbidities significantly impacts transition probabilities more than individual conditions like diabetes. It indicates that a comprehensive evaluation of multiple comorbidities is crucial for understanding patient trajectories. These findings emphasize the need for targeted strategies to manage high-risk groups with multiple comorbidities, focusing on enhancing monitoring and timely interventions. Future work will integrate the developed code into the hospital’s Health Information System (HIS) following ISO 13606 electronic health record (EHR) standards [44], with plans to implement a recursive method to enhance further the model’s efficiency and accuracy in predicting hospital length of stay and patient transitions.

## Figures and Tables

**Figure 1 life-14-01195-f001:**
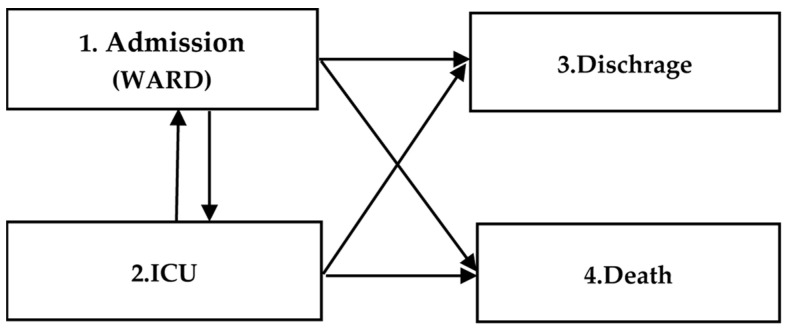
A 4-state model based on transitions between different states.

**Figure 2 life-14-01195-f002:**
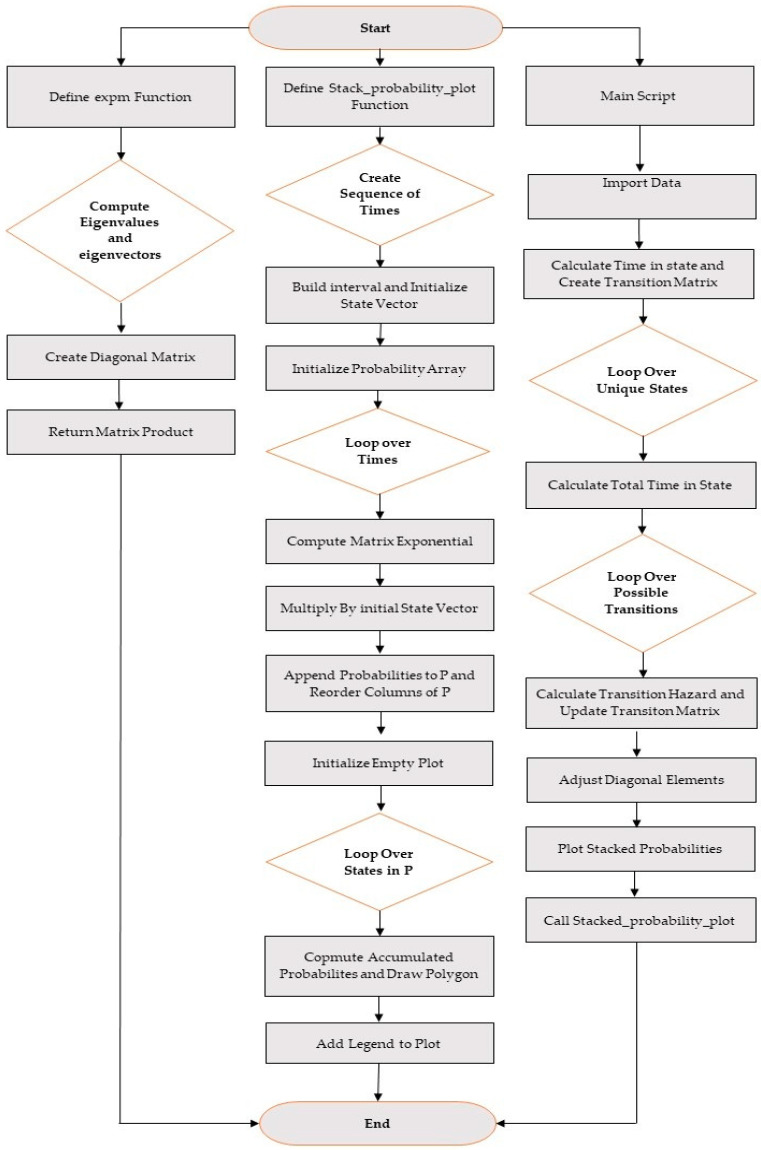
Flowchart related to programming code.

**Figure 3 life-14-01195-f003:**
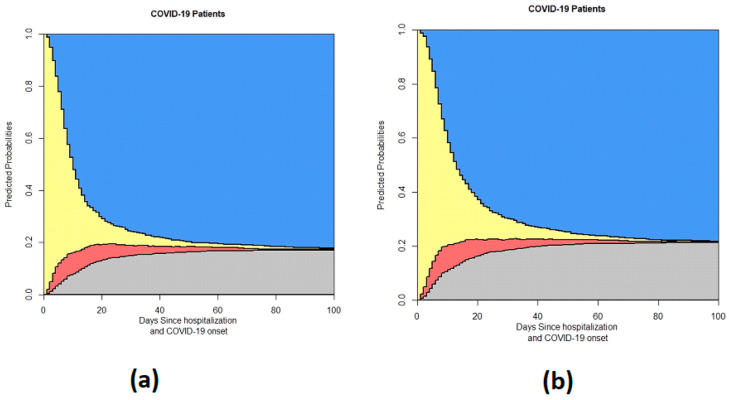
This figure shows the output of the 4-state model based on symptom duration: (**a**) symptom duration > 7; (**b**) symptom duration < 7. In the stacked probability plot, yellow, red, gray, and blue correspond to admission (Ward), ICU, death, and discharge.

**Figure 4 life-14-01195-f004:**
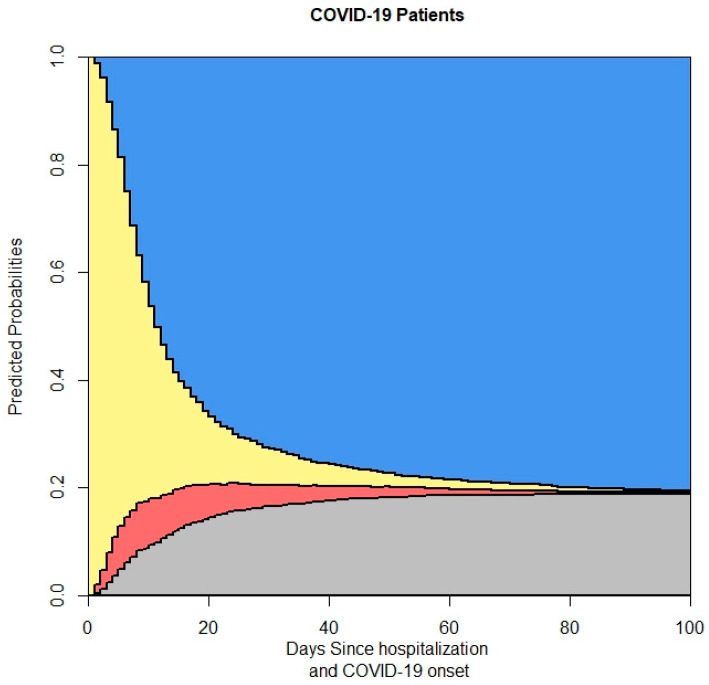
This figure shows the output of the 4-state model. In the stacked probability plot, yellow, red, gray, and blue correspond with admission (Ward), ICU, death, and discharge.

**Figure 5 life-14-01195-f005:**
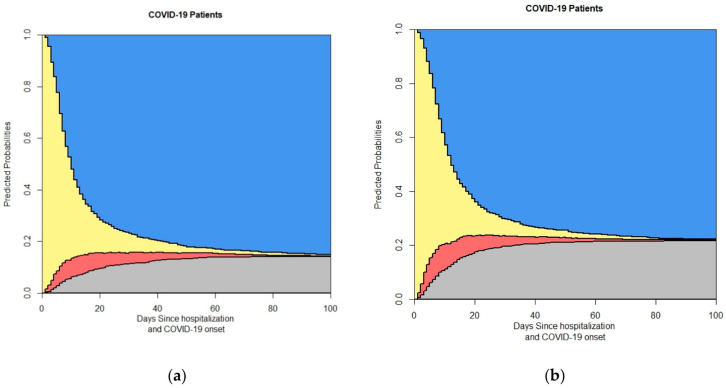
This figure shows the output of the 4-state model based on sex: (**a**) Female; (**b**) Male. In the stacked probability plot, yellow, red, gray, and blue correspond with admission (Ward), ICU, death, and discharge.

**Figure 6 life-14-01195-f006:**
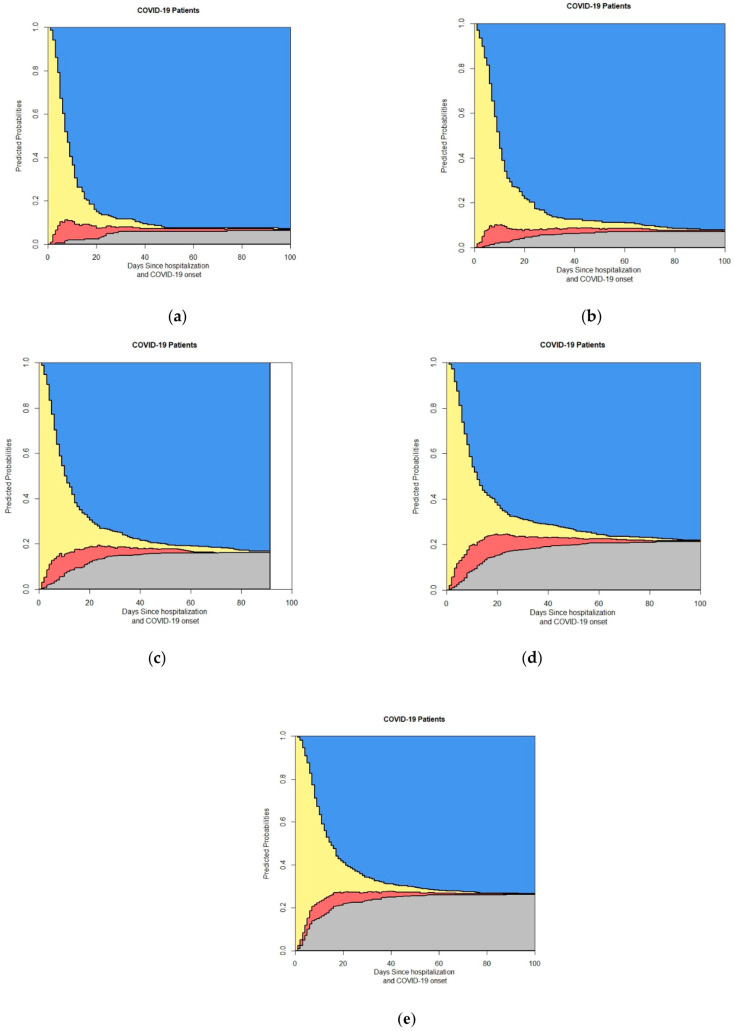
This figure shows the output of the 4-state model with age grouping. This grouping includes the following items: (**a**) Age group less than 45 years (B0); (**b**) B1 is age group between 45 and 54 years (45–54); (**c**) B2 is age group between 55 and 64 years (55–64); (**d**) B3 is age group between 65 and 74 years (65–74); (**e**) B4 is Age group more than 75 years. In the stacked probability plot, yellow, red, gray, and blue correspond with admission (Ward), ICU, death, and discharge.

**Figure 7 life-14-01195-f007:**
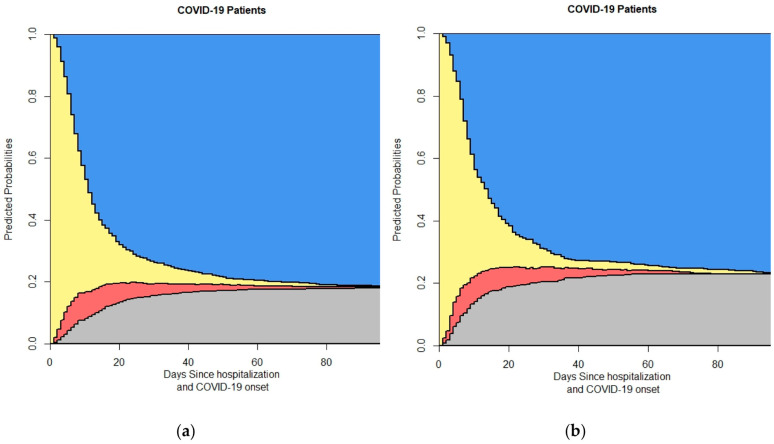
This figure shows the output of the 4-state model based on people with diabetes and without diabetes: (**a**) non-diabetic people; (**b**) diabetic people. In the stacked probability plot, yellow, red, gray, and blue correspond with admission (Ward), ICU, death, and discharge.

**Figure 8 life-14-01195-f008:**
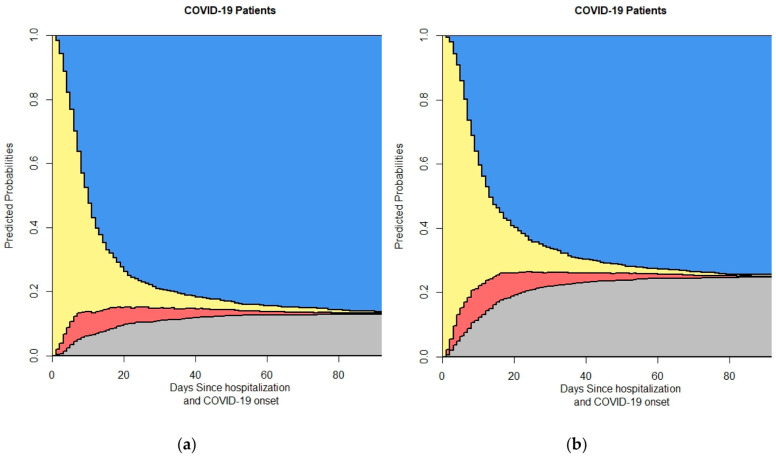
This figure shows the output of the 4-state model with the grouping of lymphocytes: (**a**) With lymphocytes > 910 × 10^6^/L (**b**) With lymphocytes < 910 × 10^6^/L. In the stacked probability plot, yellow, red, gray, and blue correspond with admission (Ward), ICU, death, and discharge.

**Figure 9 life-14-01195-f009:**
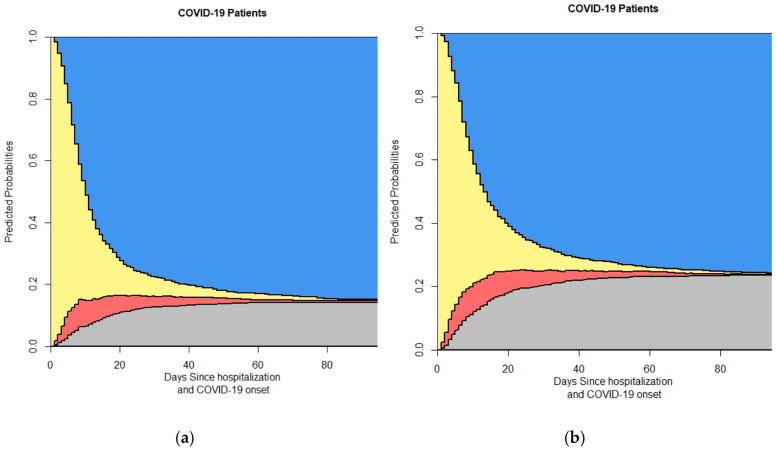
This figure shows the output of the 4-state model with the grouping of CCI: (**a**) With CCI = 0 (**b**) With CCI > 0. In the stacked probability plot, yellow, red, gray, and blue correspond with admission (Ward), ICU, death, and discharge.

**Figure 10 life-14-01195-f010:**
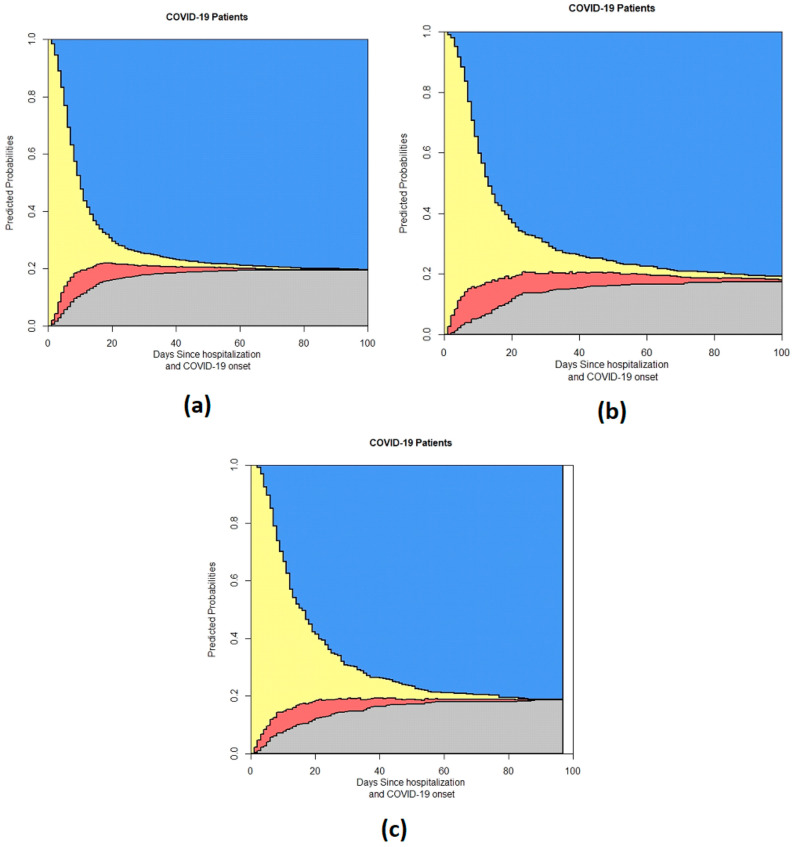
Output of the 4-state model grouped by COVID-19 waves: (**a**) First wave, (**b**) Second wave, (**c**) Third wave. In the stacked probability plot, yellow represents admission (Ward), red represents ICU, gray represents death, and blue represents discharge.

**Table 1 life-14-01195-t001:** Descriptive statistics of patient characteristics and outcomes, comparing discharged patients and those who experienced in-hospital mortality.

	Death	Discharged	Overall	*p*-Value
	(N = 437)	(N = 1848)	(N = 2285)	
**Age**				
Mean (SD)	74.8 (10.8)	64.0 (14.9)	66.1 (14.8)	<0.001
Median [Min, Max]	75.5 [42, 100]	64.7 [18, 103]	67.6 [18, 103]	
**Gender**				
Male	311 (71.2%)	1110 (60.1%)	1421 (62.2%)	<0.001
Female	126 (28.8%)	738 (39.9%)	864 (37.8%)	
**Diabetes**				
Non-Diabetic	326 (74.6%)	1524 (82.5%)	1850 (81.0%)	<0.001
Diabetes	111 (25.4%)	324 (17.5%)	435 (19.0%)	
**Lymphocyte (×10^6^/L)**				
Mean (SD)	945 (1670)	1100 (1750)	1070 (1730)	0.0809
Median [Min, Max]	750 [0, 30,800]	960 [20.0, 66,100]	910 [0, 66,100]	
Missing	6 (1.4%)	12 (0.6%)	18 (0.8%)	
**CCI**				
0	134 (30.7%)	1020 (55.2%)	1154 (50.5%)	<0.001
>0	303 (69.03%)	828 (44.8%)	1131 (49.5%)	

CCI: Charlson Comorbidity Index.

## Data Availability

The data are not publicly available due to confidentiality agreements and privacy concerns but can be accessed upon reasonable request to ensure proper use and adherence to ethical guidelines. Additionally, we have provided a representative dataset to allow testing of our code.

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
