# Peer review of "Tracing In-Hospital COVID-19 Outcomes: A Multistate Model Exploration (TRACE)"

_life, 2024, doi:10.3390/life14091195_

Round 1
Reviewer 1 Report
Comments and Suggestions for Authors
This is an intersting paper regarding a multistate model exploring COVID- 19 outcomes. In my opinion a few points have clearly to be addresed.
1. The introduction is too long ,approximately 3.5 pages. It has to be significantly reduced to approximately page max. Figure 1 should be removed. The heading of Introduction must not stand alone on the last line f the page but rather be in the next page.
2. In Figure 2 it is not clear what does the arrow going from ICU to admission mean. Please clarify.
3.More details should be given about the probabilities given since their abbreviated symbols in the tables do not provide information i.e p12, p113, q12 etc...have to be clarified. In addition the data in the matrices should be clarified.
4. Paragraph 2.3 regarding the subgroup analysis should be incorporated in the statistical analysis paragraph in 2.5. Paragraph 2.4 regarding the Jouden index should be deleted and a brief sentence incorporated in the statistical analysis paragraph i.e. in 2.5. The ROC, specificity and sensitivity temrs are too well known and need no elaboration.
5. The word lymphoma must be removed form the relelvant figure addresing the lymphocyte number and COVID-19 outcome.
6. It seems that in Table 3 the data pertaing to the Charlson Comorbities Index is not adding up to the number of patients in the column. i.e 240 patients are mentioned in the column describing the dead patients instead of 437 total. Are data missing? If so it has to be clarified.
7. 106 should be modified to 106 in manuscript and tables.
8. The author have to clarified the sentence regarding the definition of Partial Oxygen Pressure less than 300 mmHg in lines 213 -214. What do they mean? [robable they have to delete the words ''defined as''
9. A note from the authors regarding the specfic SARS-CoV-2 variant has to be added in order to specify for which variant their data are referring to.
Comments on the Quality of English LanguageGood quality of English.
Reviewer 2 Report
Comments and Suggestions for Authors
I would like to congratulate the authors their scientific idea to fill the detected knowledge gap. I read the manuscript with pleasure and high interest. This paper requires mainly editorial improvement before the next steps of publication.
Here are my major concerns:
1. Figures 4-9 can be better described to help future readers understand the crucial outcome of this research. It took me a while to interpret the "yellow" state labeled as "admission". How is it possible that a small proportion of patients persists in "admission" state? It is rather "uneventful hospitalization". Furthermore I feel lack of statistical confirmation of the presented results. I did not find Youden index in any of the presented models despite its description in the "Methods" section. Please elaborate on these figures and their legends as it is the most interesting part of the manuscript and has to be clearer.
2. I feel confused with the part of Introduction starting in line 92. I do understand that the authors wanted to share their literature review which led them to construct their multi-state model. However, listing a number of papers without broader comment does not lead a future reader to the awareness why the selected manuscript were so important for the authors. I suggest rethinking this part of your paper. After rewriting this paragraph, I would consider if it is more suitable for Background rather than Discussion?
3. I do not feel convinced with the Discussion section that everything was said what was supposed to be here. There is no paragraph about limitations of this study. From my standpoint the most essential is discrepancy between the disease onset and the admission to hospital. There was definitely a wide spectrum of duration of symptoms before coming to hospital, presumably 1 to 7 days. It may have influenced the results and it has to be discussed here.
4. I am glad to see that the authors tried to give a brief take-home message in the "Conclusions" section. I would further improve the most crucial sentence in lines 499-500. I suggest you put in brackets the worse prognostic factors e.g. gender (male), diagnosis of diabetes (not status) etc.
There are also a few minor issues:
a) lines 160-162 Please provide source for the correlations listed here.
b) lines 190-192 cite three locations of studies but only one reference [32]
c) Table 1 is rather a legend for lines 341-343, it should be deleted and the abbreviations should be defined in the text.
d) There is no Table 2 (wrong number of Table 3).
e) COVID-19 name was not introduced before line 57 (authors called it corona disease), so it requires its definition.
I am looking forward to the next round of the review.
Comments on the Quality of English LanguageThere are many sentences that are not written in good English and they unnecessarily decrease the scientific soundness of this paper. I listed a few examples below. Meticulous review of a native speaker is inevitable here.
line 130: They aimed is to determine
line 142: Hospitalized COVID-19 patients A fully parametric multistate model in a Bayesian framework
line 180: Countries with a decentralized health system will have many problems in this epidemic
Round 2
Reviewer 1 Report
Comments and Suggestions for Authors
The authors have addresed my points sufficiently.
Reviewer 2 Report
Comments and Suggestions for Authors
I am satisfied with the improvements implemented by the authors.
Comments on the Quality of English LanguageThe quality of English language has been noticeably improved.